# A bulky glycocalyx fosters metastasis formation by promoting G1 cell cycle progression

Elliot C Woods[1], FuiBoon Kai[2], J Matthew Barnes[2], Kayvon Pedram[1], Michael W Pickup[2], Michael J Hollander[1], Valerie M Weaver[2,3,4,5,6,7], Carolyn R Bertozzi[1,8]*

[1]Department of Chemistry, Stanford University, California, United States; [2]Center for Bioengineering and Tissue Regeneration, Department of Surgery, University of California, San Francisco, San Francisco, United States; [3]Department of Anatomy, University of California, San Francisco, San Francisco, United States; [4]Department of Bioengineering and Therapeutic Sciences, University of California, San Francisco, San Francisco, United States; [5]Department of Radiation Oncology, University of California, San Francisco, San Francisco, United States; [6]Eli and Edythe Broad Center of Regeneration Medicine and Stem Cell Research, University of California, San Francisco , San Francisco, United States; [7]UCSF Helen Diller Comprehensive Cancer Center, University of California, San Francisco, San Francisco, United States; [8]Howard Hughes Medical Institute, Stanford University, California, United States

**Abstract** Metastasis depends upon cancer cell growth and survival within the metastatic niche. Tumors which remodel their glycocalyces, by overexpressing bulky glycoproteins like mucins, exhibit a higher predisposition to metastasize, but the role of mucins in oncogenesis remains poorly understood. Here we report that a bulky glycocalyx promotes the expansion of disseminated tumor cells in vivo by fostering integrin adhesion assembly to permit G1 cell cycle progression. We engineered tumor cells to display glycocalyces of various thicknesses by coating them with synthetic mucin-mimetic glycopolymers. Cells adorned with longer glycopolymers showed increased metastatic potential, enhanced cell cycle progression, and greater levels of integrin-FAK mechanosignaling and Akt signaling in a syngeneic mouse model of metastasis. These effects were mirrored by expression of the ectodomain of cancer-associated mucin MUC1. These findings functionally link mucinous proteins with tumor aggression, and offer a new view of the cancer glycocalyx as a major driver of disease progression.
DOI: https://doi.org/10.7554/eLife.25752.001

*For correspondence:
bertozzi@stanford.edu

Competing interests: The authors declare that no competing interests exist.

## Introduction

Cell surface mucins, such as MUC1 and MUC16, are so consistently upregulated in epithelial cancers that they are considered reliable biomarkers of the disease. (*Bast et al., 2009*; *Rahn et al., 2001*) Despite their importance for diagnosis and prognosis, the mechanisms by which mucins might promote malignancy remain largely speculative. (*Kufe, 2009a*) To date, the majority of studies exploring the role of mucins in determining tumor phenotype have focused on the signaling function of these molecules, which resides within their short cytoplasmic tail. Yet, a striking and defining feature of cell surface mucins is their long, densely glycosylated ectodomain, which can extend hundreds of nanometers from the plasma membrane. (*Hattrup and Gendler, 2008*) Consequently, mucins can

profoundly enhance the bulk physical properties of the glycocalyx, which is a composite of the polysaccharides and glycoproteins that project from the plasma membrane and decorate all cells.

We recently showed that the bulk physical properties of the glycocalyx can exert profound effects on cellular behavior. These effects include a reorganization of cell surface receptors such as integrins. This reorganization alters their activation state and influences not only their ability to interact with extracellular matrix (ECM) ligands but also their synergistic downstream signaling with growth factor receptors. (*Paszek et al., 2014*; *Freeman et al., 2016*) By occluding the majority of binding sites, an expansive glycocalyx paradoxically promotes integrin clustering by creating a kinetic funnel in which integrins are most likely to bind to the ECM where bonds are already formed. (*Paszek et al., 2009*) In this regard, in vitro studies revealed that increasing the bulkiness of the glycocalyx enhanced the ability of nonmalignant mammary epithelial cells to grow and survive under minimally adhesive conditions whereas otherwise the cells underwent anoikis. (*Paszek et al., 2014*; *Woods et al., 2015*; *Desgrosellier and Cheresh, 2010*) Whether a bulky glycocalyx similarly modulates cell growth and survival in vivo, however, has yet to be unambiguously tested.

Importantly, we and others have also noted that tumor aggression is frequently associated with an increased expression of mucins which enhance the bulkiness of the glycocalyx and, similarly, that circulating epithelial tumor cells typically express an abundance of mucins. (*Paszek et al., 2014*) Concomitantly, primary tumor xenografts that overexpress MUC1 grow and metastasize more aggressively (*Wang et al., 2015*) and mice deficient in MUC1 resist formation of spontaneous tumors (*Spicer et al., 1995*). Together, these observations raise the intriguing possibility that tumor associated mucin expression may foster tumor progression and aggression by enhancing either tumor cell growth and survival or both by modulating integrin and growth factor receptor signaling.

Herein, we report that increasing the thickness of the glycocalyx promotes tumor metastasis by permitting cell cycle progression to facilitate cell proliferation at the metastatic site. We modulated glycocalyx thickness both genetically through ectopic expression of a tailless (signaling defective) MUC1 and more definitively by coating cells with synthetic glycopolymers with long-term cell surface retention that were designed to emulate the structure of mucin ectodomains (*Figure 1A*). (*Woods et al., 2015*) The longevity of the synthetic glycopolymers was facilitated by developing the chemistry so that the tail terminated in a cholesterylamine lipid at one end, which effectively drives spontaneous insertion into cell membranes based on mass action and the hydrophobic effect. We previously found that this particular lipid confers a long cell surface residence time to glycopolymer conjugates by mediating continuous recycling from an internal reservoir. (*Woods et al., 2015*) The core of the glycopolymer comprises *N*-acetylgalactosamine (GalNAc) residues affixed to a poly (methyl vinyl ketone [MVK]) backbone via oxime linkages. The spacing afforded by this structure approximates that of GalNAc-$\alpha$-Ser/Thr clusters within native mucins. Likewise, GalNAc-modified poly (MVK) glycopolymers emulate the physical characteristics of native mucins such as length and stiffness. (*Paszek et al., 2014*) However, unlike the more complex glycans found in native mucins, which can engage in biochemical interactions, (*Hudak et al., 2014*) the GalNAc residues we used here are considered biochemically inert in mammalian systems. Thus, these synthetic structures allowed us to model mucins' biophysical contribution to the glycocalyx without any confounding contributions from their cytosolic or extracellular biochemical activity. As well, the living polymerization chemistry used to generate the glycodomains enabled precise control of polymer length with high homogeneity. (*Godula et al., 2009*) Living polymerization techniques allow for the synthesis of polymers of very homogeneous lengths, a necessity for the comparison of the effects of long and short polymers. In this work, we used short glycopolymers of ~3 nm in length (degree of polymerization (DP) = 36 monomer units) and long glycopolymers that were ~90 nm in length (DP = 719 monomer units). As a point of comparison, integrin heterodimer pairs extend roughly 20 nm from the cell membrane (*Figure 1A*). (*Eng et al., 2011*)

## Results

Metastasis is a multistep process that requires cancer cells to overcome many checks to their irregular growth – limited not only by ability to disseminate but also to survive and thereafter to grow and expand at the metastatic site. The nonmalignant mammary epithelial cell line, MCF-10A, offers a unique model for oncogenic progression. (*Miller et al., 1993*) These cells, though immortal, lack the ability to grow or survive efficiently in a soft matrix, and therefore serve as an ideal cell line to

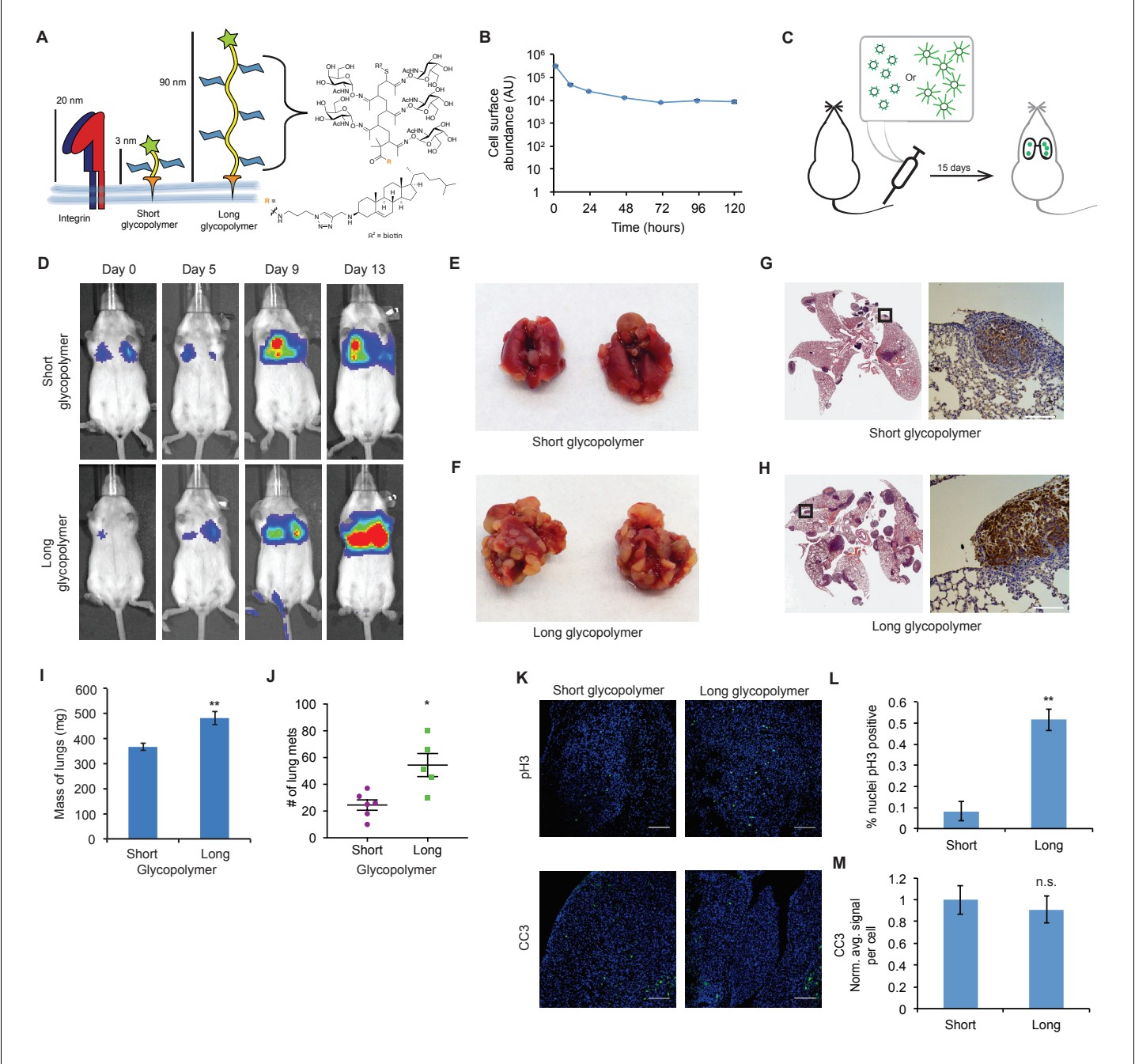

**Figure 1.** The glycocalyx increases metastatic potential in a size dependent manner. (A) Mucin-mimetic glycopolymers consist of a poly(methyl vinyl ketone) polymer with pendant GalNAc residues. The glycopolymer terminates in a synthetic sterol for insertion into cell membranes. Precise synthetic control allows for glycopolymers to be made much larger or smaller than integrins. (B) Glycopolymers reside on 4TO7 cell surfaces for days. (C) Experimental scheme for *Figure 1*. Balb/c mice were injected with syngeneic 4TO7 mammary carcinoma cells bearing long or short glycopolymers. Tail vein injections lead cells to the capillary beds of the lungs where subsequent metastatic growth can be observed. (D) Bioluminescence Imaging (BLI) of mice injected with either long or short glycopolymer-bearing cells. (E) and (F) Gross appearance of lungs excised from mice at 15 days post injection. (G) and (H) H and E staining and IHC labeling for mApple indicating presence of exogenously introduced cells. (I) Lungs excised during necropsy were weighed, wet, before fixing. (J) Frank mets were counted on whole lung sections at 5 mm into sectioning for each mouse and grouped according to glycopolymer treatment type of injected cells. (K) IF staining of mets for the mitotic marker pH3 or apoptotic marker CC3 in green. DAPI nuclear staining is shown in blue. (L) and (M) Quantification of IF signal. For pH3, shown is the quotient of positive over total nuclei. For CC3, total signal was normalized to the average signal per nuclei in mets from short glycopolymer treated cells. For both L and M, shown is the mean ±SEM of three mice per group from which 3–4 tumors were averaged each. For I and J, shown is the mean ±SEM of n = 6 for mice injected with short glycopolymer-treated cells and n = 5 for mice injected with long glycopolymer-treated cells. Scale bars are 100 μm. *p<0.05, **p<0.01 (Student's paired *t*-test).

*Figure 1 continued*

DOI: https://doi.org/10.7554/eLife.25752.002

The following figure supplements are available for figure 1:

**Figure supplement 1.** Long glycopolymers delay MCF-10A cell death in nude mice.

DOI: https://doi.org/10.7554/eLife.25752.003

**Figure supplement 2.** Long glycopolymer-bearing 4TO7 cells proliferate more rapidly in mouse lungs than those bearing short glycopolymers.

DOI: https://doi.org/10.7554/eLife.25752.004

**Figure supplement 3.** Short glycopolymer-bearing 4TO7 cells have no metastatic advantage over naïve 4TO7 cells.

DOI: https://doi.org/10.7554/eLife.25752.005

evaluate the effects of the glycocalyx on mammary epithelial cell behavior in the context of a metastatic site. (*Paszek et al., 2005*) Our prior studies suggested that a bulky glycocalyx permitted not only the survival but also the growth of these mammary epithelial cells in an in vitro model of a minimally adhesive microenvironment. Thus, we asked whether a bulky glycocalyx would permit both the survival and the growth and expansion of disseminated mammary epithelial cells in vivo. MCF-10A cells were painted with our glycopolymers, then injected into the venous system of nude mice, where they accumulated in the lungs regardless of glycopolymer length. We found that those cells treated with long glycopolymer mucin-mimetics survived longer, as demonstrated by delayed activation of the apoptosis marker cleaved caspase-3 (CC3), than those decorated with short glycopolymers (*Figure 1—figure supplement 1*), consistent with our in vitro studies. Nevertheless, the cells eventually died and therefore ultimately failed to form metastatic lesions, leaving us unable to determine the potential effect of the glycocalyx on proliferation. Thus, to unambiguously address whether or not a bulky glycocalyx might influence tumor behavior and metastasis by regulating cell proliferation, independent of cell survival, we assessed the influence of the glycopolymers on the growth and metastatic efficiency of the dormant-prone tumor cell line from the syngeneic 4TO7/Balbc model.

The murine cell line 4TO7 is a Balb/c syngeneic mammary carcinoma that is able to efficiently disseminate and survive for prolonged periods of time at metastatic sites such as the lung. (*Aslakson and Miller, 1992*) However, the tumor cells in this model form predominately dormant tumors because they are unable to proliferate efficiently at the target site. These 4TO7 cells, therefore, provided us the ideal model to specifically address the contribution of a bulky glycocalyx on cell proliferation and G1 cell cycle transit in vivo.

## The glycocalyx increases metastatic potential in a size dependent manner

We first confirmed that the mucin-mimetic glycopolymers were long-lived on 4TO7 cell surfaces in culture, as previously found with other cell lines (*Figure 1B*). (*Woods et al., 2015*) Next, we grew mApple-luciferase fusion-transfected 4TO7 cells to subconfluency, painted them with glycopolymers, then injected them into the tail veins of Balb/c mice (*Figure 1C*). Veins lead directly to the right heart where cells next meet the capillary beds of the lungs and are lodged. Every 4–5 days mice were anesthetized and given an intraperitoneal (IP) injection of luciferin for bioluminescence imaging (BLI) (*Figure 1D*). By day 13, mice injected with 4TO7 cells coated with long glycopolymers were generating greater photon flux, on average, than their counterparts injected with short glycopolymer-treated cells, suggesting a higher density of luciferase-expressing cells and thus an increased tumor burden (*Figure 1D* and *Figure 1—figure supplement 2*). In particular, we noted that the increase in BLI over time in the long glycopolymer group appeared to approximate an exponential growth-type trend, while the short group did not (*Figure 1—figure supplement 2*). Importantly, we measured no significant difference in accumulation of tumor cells in the lungs between treatment groups as measured by BLI at Day 0.

On day 15, the mice were sacrificed and necropsies were performed. We observed gross differences in tumor burden between glycopolymer treatment groups (compare *Figure 1E and F*). Lungs were weighed wet then fixed in 4% paraformaldehyde and embedded in paraffin wax for whole histological sectioning. Hematoxylin and eosin (H and E) staining revealed qualitative differences in tumor burden, and immunohistochemistry (IHC) staining for mApple confirmed that tumors were

derived from exogenous mApple-luciferase transfected 4TO7 cells and did not arise de novo (*Figure 1G and H*).

Lung tissue is spongy and consists of an abundance of negative space that can be filled with air during respiration. Tumors, by comparison, are dense and contain relatively little negative space. Consequently, in a vein-to-lung model of metastasis, lung mass can be a strong indicator of tumor burden. We found that the lungs of mice injected with 4TO7 cells endowed with a bulky glycocalyx were significantly heavier than lungs from mice injected with short glycopolymer-bearing cells (*Figure 1I*). In addition to mass, we directly quantitated the number of mets per lung. For each mouse, we counted every met in the whole lung H and E section taken five millimeters (mm) into sectioning. Long glycopolymer-treated 4TO7 cells produced, on average, more than twice the number of mets per mouse when compared to short (*Figure 1J*). Thus, merely increasing the thickness of the glycocalyx was sufficient to increase the metastatic potential of otherwise poorly metastasizing murine carcinoma cells. Importantly, the short glycopolymers did not increase the metastatic potential of 4TO7 cells when compared to mock (PBS) treated cells (*Figure 1—figure supplement 3*). Metastasis is an inefficient process—few circulating tumor cells actually give rise to metastatic tumors—and the prevailing view is that survival and proliferation at the secondary site is rate limiting.(*Chambers et al., 2002*) Therefore, we wondered if this increased metastatic potential could be due to increased growth, survival, or both.

Quantitative immunofluorescence (IF) imaging of phosphorylated histone H3 (pH3)—a marker of mitosis—and cleaved caspase-3 (CC3)—a marker of apoptosis—revealed that mets from long glycopolymer-treated cells had higher rates of proliferation than those from cells treated with short glycopolymers (*Figure 1K and L*). But, the rates of apoptosis were similar (*Figure 1K and M*). While surprising at first, given our knowledge that a thick glycocalyx improves survival in minimal adhesion settings, we reasoned that by day 15 the survival phenotype had likely run its course—cells unable to survive in that niche may have died days earlier. These data corroborated our BLI results and indicated that the glycocalyx likely affects metastatic proliferative competency in vivo.

## A thick glycocalyx drives cell cycle progression via the PI3K-Akt axis

To gain mechanistic insight into this apparent proliferative effect, we tested the effects of modulating glycocalyx thickness on cell growth in a soft matrix in vitro model. We used fibronectin-functionalized polyacrylamide (PA) gels whose stiffness can be controlled by varying both acrylamide concentration and crosslinking frequency. (*Lakins et al., 2012*) We painted MCF-10A cells with short or long mucin-mimetic glycopolymers, or with vehicle (PBS), then plated them at low density on fibronectin-functionalized soft PA gels (Young's modulus = 400 Pa) so that nearly all cells were isolated, minimizing cell-cell interactions. After 72 hr, we imaged the gels and quantified the number of cells in each observed colony. The long glycopolymer-painted cells formed colonies with, on average, twice as many cells as colonies formed by the short glycopolymer-treated cells or PBS-treated controls (*Figure 2A–C*). And the percentage of colonies growing beyond two cells nearly tripled for long-glycopolymer-treated cells verses PBS-treated controls, indicating that this increase in mean colony size was not due merely to a few large colonies (*Figure 2D*).

To test that this phenotype required cell-surface residence of the long glycopolymers, we compared the effects of the cholesterylamine-anchored molecules, which persist on the plasma membrane for several days even after cell division, with those of a phospholipid-functionalized glycopolymer that is lost from the cell surface within several hours. (*Woods et al., 2015*) Cells treated with these short-lived but equally long glycopolymers (DPPE) did not produce larger colonies compared to PBS treatment alone (*Figure 2—figure supplement 1*). Thus, the effects measured in *Figure 2C and D* are likely mediated by glycopolymers resident on the cell surface.

To further test our hypothesis that a bulky glycocalyx permits proliferation in the metastatic niche, we analyzed markers of cell cycle progression. Cells were painted with short or long glycopolymers, or with vehicle (PBS), then plated on either soft (Young's modulus = 400 Pa) or stiff (Young's modulus = 60,000 Pa) fibronectin-functionalized PA gels, the latter being a positive control for integrin activation. (*Discher et al., 2005*) After allowing them to adhere for 6 hr, cells were lysed and the lysates analyzed by immunoblotting. We found that MCF-10A cells plated on stiff PA gels, which offer a rich adhesion setting, produced higher levels of cyclin D (a protein required for cell cycle progression) and showed higher levels of pH3 than those on adhesion-poor soft gels (*Figure 2E and F*). Long glycopolymer-treated cells also demonstrated increased cyclin D and pH3 levels, compared to

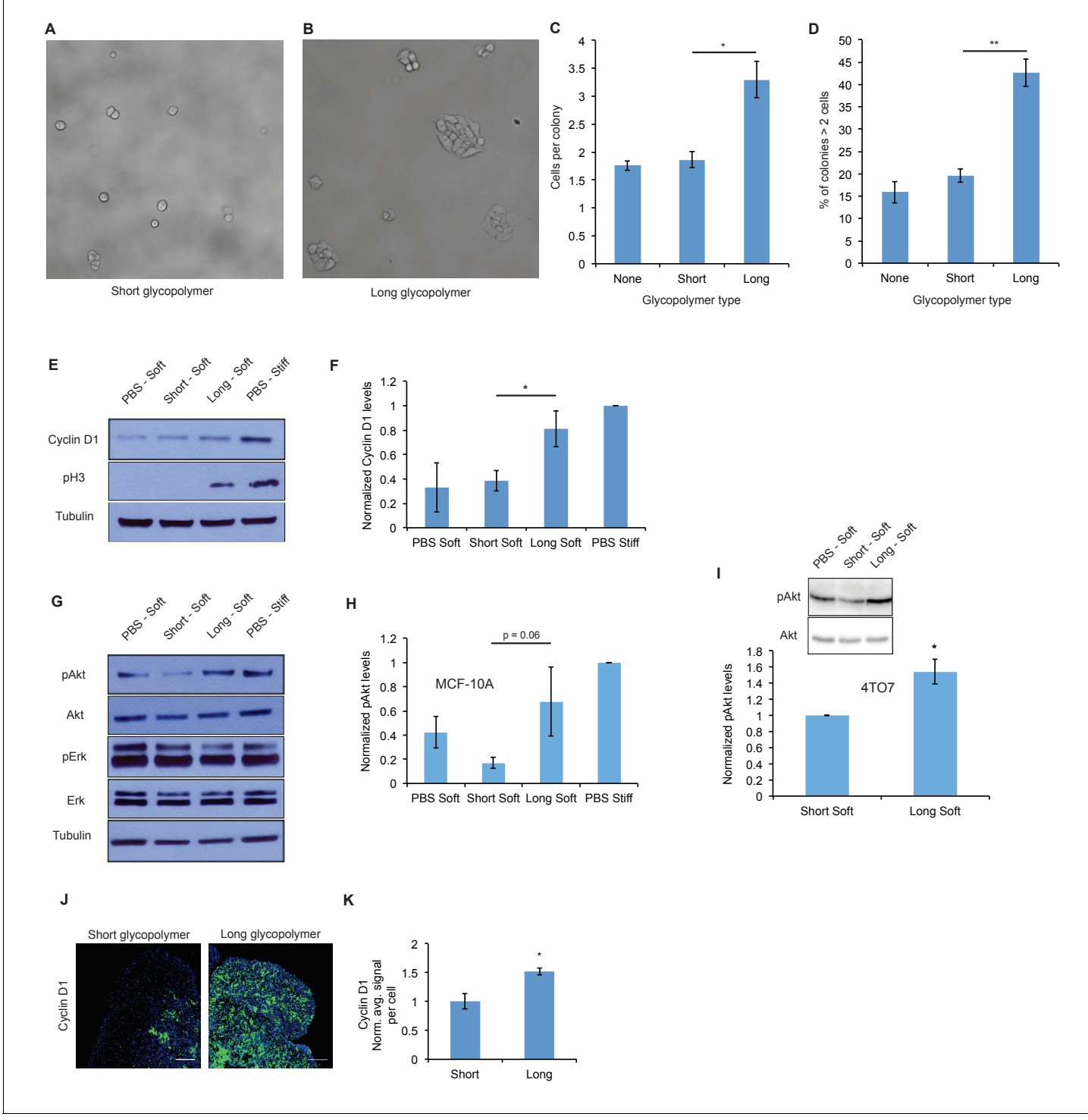

**Figure 2.** A thick glycocalyx drives cell cycle progression via the PI3K-Akt axis. (**A**) and (**B**) Microscopy of MCF-10A cells labeled with short or long glycopolymers, plated on soft (400 Pa) fibronectin-functionalized gels, and cultured for 72 hr. (**C**) Average number of cells per colony in images such as A and B. (**D**) Percent of colonies containing more than 2 cells in images such as A and B. (**E**) Immunoblot analysis of proliferative markers in MCF-10A cells coated with long or short glycopolymers or treated with vehicle (PBS) and plated on soft (400 Pa) or stiff (60 kPa) fibronectin-functionalized polyacrylamide gels for six hours. (**F**) Western blots from E were analyzed by densitometry, their values normalized first to total protein (tubulin) then to that of the positive control—PBS treated cells on a stiff Fn-functionalized matrix. (**G**) Immunoblot analysis of pAkt and pErk in MCF-10A cells serum-starved for 72 hr, treated with long or short glycopolymers or vehicle and plated in serum-free media on soft or stiff gels for six hours, then challenged with epidermal growth factor for 15 min, and lysed. (**H**) Western blots from G were quantified and normalized to total Akt then to the positive control.
*Figure 2 continued on next page*

*Figure 2 continued*

(I) Immunoblot and quantification of pAkt and total Akt in 4TO7 cells in vitro treated as in G and quantified as in H except normalized to short glycopolymer control. (J) Cyclin D1 (green) IF staining of 4TO7 mets from experiments in *Figure 1*. DAPI nuclear stain in blue. (K) Quantification of IF staining normalized to the average signal per nucleli in mets from short glycopolymer treated cells. Shown in K is the mean ±SEM of three mice per group from which 3–4 tumors were averaged each. For C, D, F, and I, shown is the mean ±SEM of three biological replicate experiments. For H, shown is the mean ±SEM of four biological replicate experiments. Scale bars are 100 μm. *p<0.05, **p<0.01 (Student's paired *t*-test).

DOI: https://doi.org/10.7554/eLife.25752.006

The following figure supplements are available for figure 2:

**Figure supplement 1.** Glycopolymers must be cell-surface resident to affect colony size.
DOI: https://doi.org/10.7554/eLife.25752.007
**Figure supplement 2.** Glycopolymers do not abolish need for growth factor signals in proliferation.
DOI: https://doi.org/10.7554/eLife.25752.008
**Figure supplement 3.** Long glycopolymers increase active Akt in 4TO7 lung metastases.
DOI: https://doi.org/10.7554/eLife.25752.009
**Figure supplement 4.** PI3K or MEK inhibitor abolishes effects of long glycopolymers on proliferation.
DOI: https://doi.org/10.7554/eLife.25752.010
**Figure supplement 5.** Cyclin D1 expression in 4TO7 cells in vitro.
DOI: https://doi.org/10.7554/eLife.25752.011

short glycopolymer- and PBS-treated cells, even on soft gels, suggesting that a thick glycocalyx drove proliferation and cell cycle progression in this poor adhesion setting.

Integrin activation and growth factor signaling form, in logic terms, an 'AND gate' for cell cycle progression in nontransformed cells. (*Kuwada and Li, 2000*) We sought to determine whether a bulky glycocalyx alters this fundamental control mechanism. After serum-starving MCF-10A cells for 72 hr, we lifted them and painted them with long or short glycopolymers, or with vehicle. The cells were plated on gels as in *Figure 2E*, except now in serum-free media, and then lysed after 6 hr. A lack of growth factor signaling reduced cyclin D1 and pH3 levels to below the detection limit, even when cells were grown on stiff gels or endowed with a bulky glycocalyx (*Figure 2—figure supplement 2*). The underlying drivers of proliferation therefore, including growth factor signaling requirements, appeared to be unaltered by glycocalyx thickness in this model.

To discern the factors responsible for increased cell cycle progression in cells with thicker glycocalyces, we examined Erk and Akt, the mitogen activated protein kinases (MAPKs) thought to be central to adhesion-mediated control of cell cycle progression. (*Moreno-Layseca and Streuli, 2014*) After serum starvation for 72 hr, cells were coated with glycopolymers or vehicle, and plated on PA gels in serum-free media for 6 hr. The cells were then challenged with epidermal growth factor (EGF) for 15 min and then lysed. A bulky glycocalyx increased Akt phosphorylation (pAkt), a proxy for activation, on soft gels relative to that in cells with short glycopolymers, while Erk phosphorylation, however, remained unchanged (*Figure 2G and H*). The same effect on pAkt was seen on 4TO7 cells in vitro (*Figure 2I*). As well, IF analysis of tissue sections from the experiments in *Figure 1* revealed increased Akt activation, as measured by staining for phosphorylated Akt substrate, in vivo in long glycopolymer-bearing tumor cells when compared to short (*Figure 2—figure supplement 3*). Consistent with this finding, we repeated the colony formation experiments and found that proliferation of long glycopolymer-painted cells was blocked by an inhibitor of phosphoinositol-3 kinase (PI3K), an upstream activator of Akt, or an inhibitor of MEK—a MAPK downstream of growth factor receptors (*Figure 2—figure supplement 4*). A bulky glycocalyx, therefore, seems to drive cell cycle progression via the PI3K-Akt axis.

While 4TO7 cells in vitro did not experience significantly increased cyclin D1 expression with a thicker glycocalyx (*Figure 2—figure supplement 5*), we were able to confirm the role of glycocalyx-driven cell cycle progression in promoting metastasis in vivo by IF staining of the 4TO7 lung mets from the experiments in *Figure 1*. Metastatic 4TO7 cells in vivo expressed significantly more of the cell cycle progression marker cyclin D1 when bearing long glycopolymers verses short (*Figure 2J and K*). We posit that transformed cells, such as 4TO7, in rich culture media in vitro may see only a negligible increase in proliferative signal from a bulky glycocalyx as evidenced by the robust cyclin D1 signal in the negative control: PBS treated 4TO7 cells on soft gels.

# A thick glycocalyx stimulates integrin-FAK mechanosignaling

Next we sought to test our hypothesis that this phenotype is driven by integrin- focal adhesion kinase (FAK) signaling. FAK disseminates adhesion information from focal adhesions to the rest of the cell via autophosphorylation at tyrosine 397, as well as increased phosphorylation at tyrosine 925, presumably through Src kinase activation. (*Mitra et al., 2005*) Western blotting with phospho-

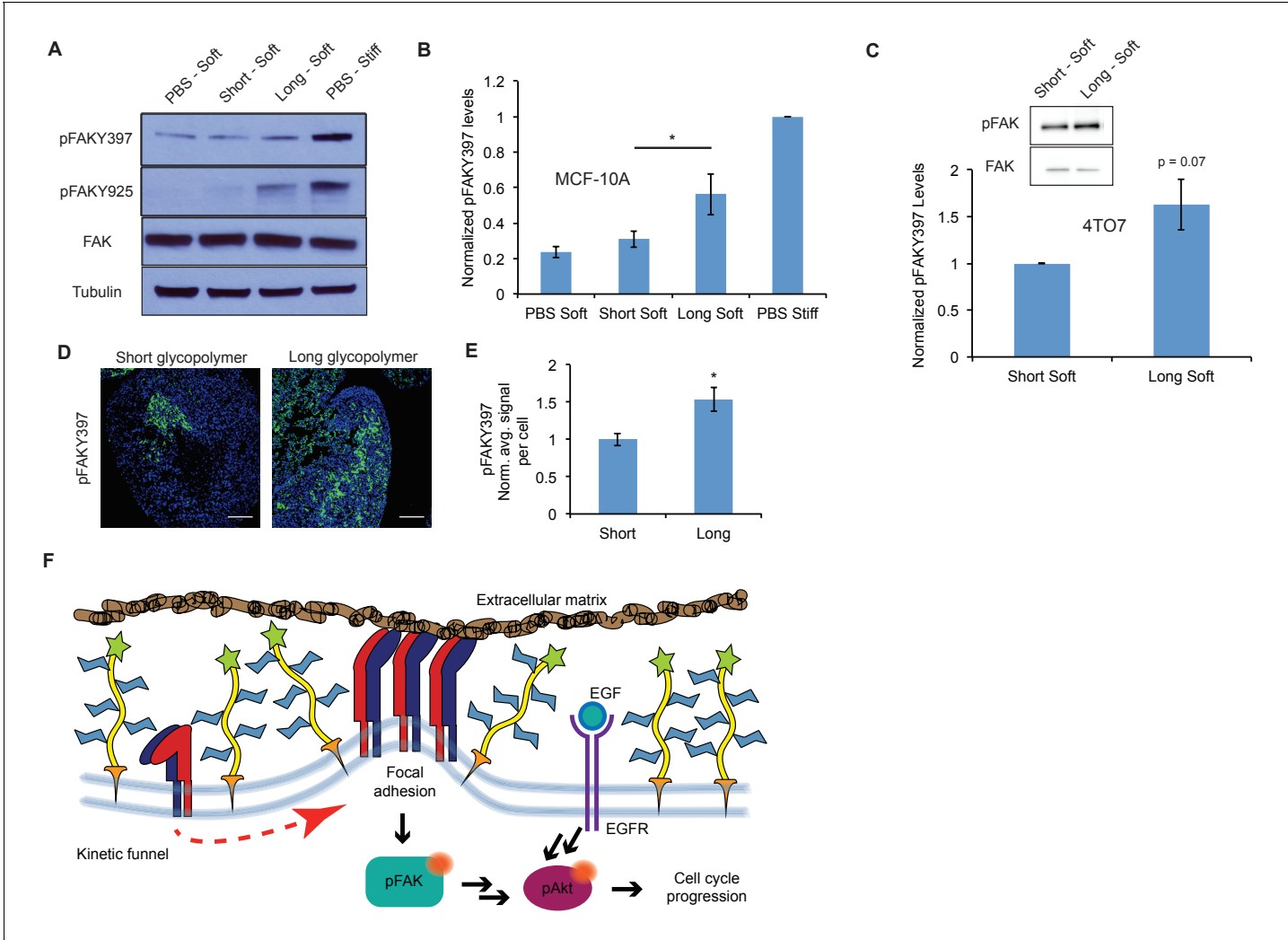

**Figure 3.** A thick glycocalyx stimulates integrin-FAK mechanosignaling. (A) Immunoblot analysis of FAK phosphorylation in MCF-10A cells coated with long or short glycopolymers or treated with vehicle (PBS) and plated on soft (400 Pa) or stiff (60 kPa) fibronectin-functionalized polyacrylamide gels for six hours. (B) Western blots from A were analyzed by densitometry, their values normalized first to total FAK then to that of the positive control—PBS treated cells on a stiff fibronectin-functionalized matrix. (C) Immunoblot analysis and quantification of pFAK and total FAK in 4TO7 cells treated as in A and quantified as in B except normalized to short glycopolymer control. D) pFAKY397 IF staining in green of mets from experiments in *Figure 1*. DAPI nuclear stain in blue. (E) Quantification of IF staining in D normalized to the average signal per nuceli in mets from short glycopolymer-treated cells. Shown is mean ±SEM of three mice per group from which 3–4 tumors were averaged each. (F) A model by which a mucin-bolstered glycocalyx may drive proliferation. Limited ligand access due to steric hindrance establishes a kinetic funnel in which integrins are likely to bind where bonds exist already. This drives a clustering of integrins that activates FAK which, in conjunction with EGFR, drives activation of Akt and subsequently cell cycle progression. Blot in A is representative of at least three biological replicate experiments. Shown in B is mean ±SEM from five biological replicate experiments. Shown in C is mean ±SEM from three biological replicate experiments. Scale bars are 100 μm. *p<0.05 (Student's paired *t*-test).
DOI: https://doi.org/10.7554/eLife.25752.012

The following figure supplement is available for figure 3:

**Figure supplement 1.** FAK inhibitor abolishes proliferative advantage from bulky glycocalyx.
DOI: https://doi.org/10.7554/eLife.25752.013

specific antibodies revealed that possession of a thick glycocalyx led to enhanced activation of FAK (*Figure 3A and B*). These effects on FAK were mirrored by adhesion of untreated cells to a stiff matrix. By contrast, cells coated with short glycopolymers, or left untreated, and plated on a soft matrix showed comparatively less phosphorylation of FAK. Similar results were seen with 4TO7 cells in vitro (*Figure 3C*). Reciprocally, cells treated with long glycopolymers along with a nonlethal dose of a specific inhibitor of FAK were unable to form colonies on soft matrices, whereas vehicle (DMSO)-treated, long glycopolymer-coated cells retained colony forming activity (*Figure 3—figure supplement 1*).

To determine whether enhancement of adhesion signaling due to a bulky glycocalyx also occurs in vivo, we probed FAK phosphorylation status in the mets from *Figure 1*. Long glycopolymer-treated cells formed tumors with significantly more FAK activation than tumors formed from short glycopolymer-bearing cells (*Figure 3D and E*). These data, taken with the results in *Figure 2*, support a model for glycocalyx-driven enhancement of proliferation in minimal-adhesion settings. A bulky glycocalyx drives integrin clustering through kinetic funneling, leading to increased signaling from focal adhesion-associated proteins such as FAK. In combination with growth factor signaling, this leads to enhanced MAPK activation, such as Akt phosphorylation, which then promotes G1 cell cycle progression through expression of proteins like the cyclins (*Figure 3F*).

## The MUC1 ectodomain is sufficient to increase the metastatic potential of 4TO7 cancer cells

Lastly, for physiological relevance, we evaluated the effects of a natural mucin ectodomain on the metastatic potential of 4TO7 cells. We utilized a cytoplasmic truncation of MUC1 (MUC1ΔCT), the mucin most commonly overexpressed in breast and ovarian cancers, to limit any cytoplasmic signaling which might confound our results. We transfected mApple-luciferase expressing 4TO7 cells with MUC1ΔCT under the control of a doxycycline (dox)-inducible promoter. We grew these cells to subconfluence, treated them with doxycycline (200 ng/ml) or vehicle (PBS) for 24 hr, then lifted them and immediately injected them into Balb/c mouse tail veins (*Figure 4A*). As previously, mice were monitored every 4–5 days via BLI measurements and sacrificed on day 15.

BLI photon flux revealed that MUC1ΔCT-expressing cells formed mets that were growing more rapidly than those formed from MUC1ΔCT-negative cells (*Figure 4—figure supplement 1*). Whole lung sectioning, H and E staining and IHC on mApple revealed a dramatic qualitative difference in tumor burden (*Figure 4B and C*). Wet lung mass was nearly doubled in mice injected with MUC1ΔCT-expressing cells versus control (*Figure 4D*). Nearly three times the number of mets were formed in the lungs of mice injected with cells expressing the mucin ectodomain compared to those that were not (*Figure 4E*). pH3 IF staining revealed that more than twice the number of nuclei in MUC1ΔCT-expressing cells were actively mitotic. And while the values did not reach statistical significance (p=0.06), CC3 levels were trending towards lower values in MUC1ΔCT-expressing mets (*Figure 4F–H*). Staining for pY397-FAK, cyclin D1 and pAkt substrate demonstrated that the MUC1 ectodomain increases mechanosignaling, cell cycle progression and MAPK activity in lung mets, further supporting our model (*Figure 4I–K* and *Figure 4—figure supplement 2*). These results dramatically demonstrate the effect that the mucin ectodomain can have on metastatic cells in vivo.

## Discussion

Mucin overexpression is commonly observed in epithelial cancers. MUC1 in particular is overexpressed in ~64% of tumors of all types diagnosed each year in the U.S. (*Kufe, 2009b*), rendering MUC1 one of the most prominently dysregulated genes in cancer. As a point of reference, Ras (*K-*, *H-* and *N-RAS* combined) mutations are estimated to occur in 9–30% of all cancers. (*Cox et al., 2014*) Hypotheses regarding the mechanism by which MUC1 overexpression drives cancer progression have focused almost entirely on biochemical interactions of its 72-residue cytosolic domain (*Kufe, 2009b*), which represents <10% of the overall protein sequence. The bulk of MUC1 resides outside the cell where it dominates the physical properties of the glycocalyx. In previous work, we showed that this ectodomain profoundly influences focal adhesion formation, integrin signaling, and survival in a minimal adhesion setting. (*Paszek et al., 2014*) But this effect alone cannot explain the striking effect of MUC1 ectodomain expression on metastatic burden that we observed in this study (*Figure 4*). Our data herein show that a bulky glycocalyx, achieved either with synthetic or natural

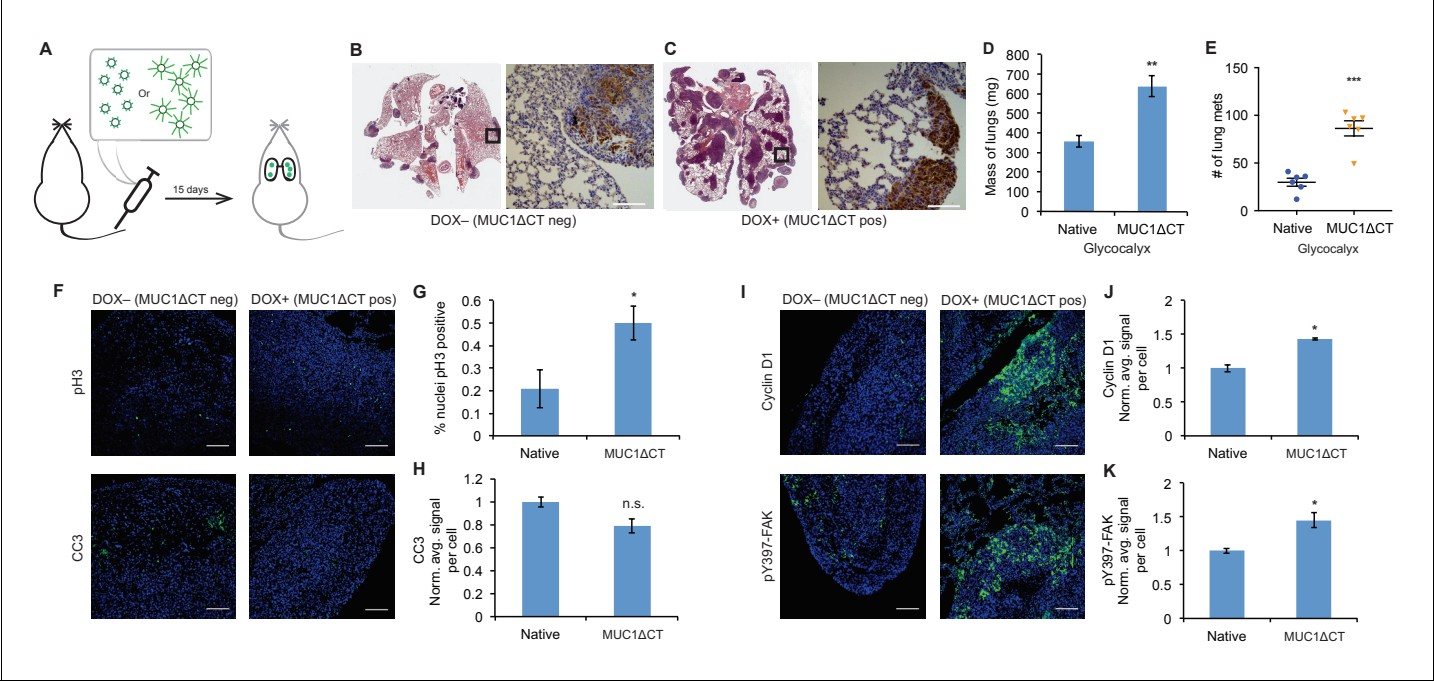

**Figure 4.** The MUC1 ectodomain is sufficient to increase the metastatic potential of 4TO7 cancer cells. (A) Experimental scheme for *Figure 4*. Balb/c mice were injected with syngeneic 4TO7 mammary carcinoma cells transfected with a MUC1ΔCT construct under the control of a doxycycline (dox)-inducible promoter. Cells were treated with either dox 200 ng/ml or vehicle (PBS) for 24 hr before injections. (B) and (C) Whole lung sections H and E stained and IHC labeled for mApple indicating presence of exogenously introduced cells. (D) Lungs excised during necropsy were weighed, wet, before fixing. (E) Frank mets were counted on whole lung sections at 5 mm into sectioning for each mouse and grouped according to treatment type of injected cells. (F) IF staining of mets for the mitotic marker pH3 or apoptotic marker CC3 in green. DAPI nuclear staining is shown in blue. (G) and (H) Quantification of IF signal. For pH3, shown is the quotient of positive over total nuclei. (I) Cyclin D1 and pY397-FAK IF staining of mets in green. DAPI nuclear stain in blue. (J) and (K) Quantification of IF signal. For H, J, and K, total signal was normalized to the average signal per nuceli in mets from PBS treated cells. Shown is mean ±SEM of three mice per group from which 3–4 tumors were averaged each. For D and E, shown is the mean ±SEM of n = 6 for mice injected with PBS-treated cells and n = 6 for mice injected with dox-treated cells. Scale bars are 100 μm. *p<0.05, **p<0.01, ***p<0.001 (Student's paired *t*-test).

DOI: https://doi.org/10.7554/eLife.25752.014

The following figure supplements are available for figure 4:

**Figure supplement 1.** 4TO7 cells expressing the MUC1 ectodomain proliferate more rapidly in mouse lungs than those that do not.
DOI: https://doi.org/10.7554/eLife.25752.015

**Figure supplement 2.** The MUC1 ectodomain increases Akt activity in 4TO7 lung metastases.
DOI: https://doi.org/10.7554/eLife.25752.016

mucins, also promotes proliferation in the metastatic niche. The mucin ectodomain promotes mechanosignaling and enhances cell cycle progression via the PI3K-Akt axis. This model unifies the structure of mucins with their consistent overexpression in metastatic disease (*Horm and Schroeder, 2013*) and the correlation of their overexpression with poor prognosis. (*Rahn et al., 2001*; *Duffy et al., 2000*; *Retterspitz et al., 2010*; *McGuckin et al., 1995*) As well, importantly, our results imply that drugs targeting the cytoplasmic tail of MUC1 will be missing a key pathophysiologic mechanism.

It should be noted that in addition to their physical influence, the glycans on mucins have been found to participate in biochemical interactions. For example, sialylated mucin-associated glycans engage the Siglec family of immunomodulatory receptors and may therefore tune the response of critical effector cells in the tumor microenvironment. (*Belisle et al., 2010*; *Ohta et al., 2010*; *Beatson et al., 2016*) Thus, mucins' influence on cancer likely reflects many functional modalities, each contributing differentially to various facets of disease progression. From this vantage point, mucins are prime targets for therapeutic intervention and warrant increased focus on avenues for their disruption.

## Materials and methods

### Mucin-mimetic glycopolymers

Glycopolymers were synthesized as previously described (*Woods et al., 2015*). Briefly, lipid-conjugated RAFT agents were synthesized, from which methyl vinyl ketone was polymerized, to generate polymers of various lengths with low polydispersities. The ketone pendant groups were functionalized with alkoxy-amine containing *N*-acetylgalactosamine monomers and purified to give the final lipid-terminated glycopolymers.

### Cell culture

MCF-10As were acquired from ATCC (CLR-10317), met the published morphological characteristics described on ATCC, used within 30 passages, confirmed to be mycoplasma free, and maintained in recommended media: DMEM/F12 without phenol red, supplemented to contain 5% horse serum (Invitrogen, Carlsbad, CA), EGF (20 ng/ml from Peprotech, Rock Hill, NJ), hydrocortisone (0.5 mg/ml from Sigma, St. Louis, MO), cholera toxin (100 ng/ml from Sigma), insulin (10 µg/ml from Sigma) and 1% penicillin/streptomycin.

4TO7 cells were a kind gift from Dr. Philip Owens (Vanderbilt University), met the morphologic characteristics described on ATCC, used within 30 passages, confirmed to be mycoplasma free and were maintained in DMEM with 10% FBS and 1% penicillin/streptomycin.

### Glycopolymer loading onto cell surfaces

Lipid-anchored glycopolymers spontaneously insert into cell membranes due to the hydrophobic effect and mass action. Where noted in experimental sections below, cells in suspension are mixed with stock concentrates of aqueous solutions of glycopolymers and allowed to be labeled for 1 hr at room temperature. The cells are then pelleted, the labeling solution aspirated, and the excess glycopolymers washed away by an additional dilution-pelleting step. Details of the synthesis and use of glycopolymers used in this study can be found in Woods *et al* (*Woods et al., 2015*).

### Cell surface measurement of glycopolymers

Glycopolymers bearing a biotin molecule on the terminus opposite the lipid tail were incubated with cells. Cells were washed and incubated in warm complete media for the desired length of time, then washed with ice-cold PBS to reduce lipid trafficking and incubated with fluorescent anti-biotin antibodies at 4°C for 20 min, then washed and analyzed by flow cytometry. More details of this method of measuring recycling of lipid-born glycopolymers can be found in Woods *et al* (*Woods et al., 2015*).

### mApple-Luciferase transfection

4TO7 cells were stably transduced with an mApple-luciferase fusion with lentivirus (pLV). Cells were selected by flow cytometry in the red channel for purity then used as described (*Yang et al., 2004*).

### MUC1ΔCT transfection

4TO7 cells were stably transduced with reverse tetracycline-controlled transactivator (rtTA, tet-on system) lentivirus (pLV-neo). After neomycin (100 µg/mL) selection of rtTA-integration, cells were co-transfected (Lipofectamine 3000, Thermo Fisher, per manufacturer's recommendation) with a transposon (PiggyBac) expressing human MUC1ΔCT (cytoplasmic tail-deleted) and PiggyBac transposase. Cells were selected in puromycin (1 µg/mL) for purity and then used as described (*Yang et al., 2004*).

### Polyacrylamide cell substrates

Fibronectin-functionalized PA substrates were synthesized as described previously with a few modifications. Briefly, methacrylate-functionalized cover glass was used with dichlorodimethylsilane-functionalized cover glass to create a sandwich between which PA gels were allowed to polymerize. Gels were functionalized with the heterobifunctional molecule N6. Fibronectin then was conjugated to the gels via N6's amine-coupling chemistry. Gels were rinsed and warmed with media before cells were added.

## Colony formation experiments

MCF-10A cells were lifted with trypsin, counted, then incubated with 10 µM polymers in PBS or PBS alone at $10^7$ cells per ml for 1 hr at room temperature. Cells were washed, then brought up into warm media and plated onto fibronectin-functionalized substrates. After 72 hr in culture, images were taken from nine random fields of view each from two technical replicate gels per experimental condition, and the experiment was replicated in entirety thrice.

## Western blots

MCF-10A cells were lifted and coated with polymers as in the colony experiments. Six hours after plating at $4.6 \times 10^3$ cells per $cm^2$, the cells were rinsed with ice-cold PBS, then lysed with M-PER lysis buffer (Thermo Fisher, Waltham, MA) containing phosphatase inhibitors and protease inhibitors. Lysates were subjected to SDS-PAGE and transferred to nitrocellulose membranes. Primary antibodies were from Cell Signaling Technology (Danvers, MA) and used as recommended. HRP-conjugated secondary antibodies were from Cell Signaling Technology and used as recommended.

## Mice

All mice were maintained in accordance with University of California Institutional Animal Care and Use Committee guidelines under protocol AN105326. Four week old female Balb/c or nude mice were obtained from Jackson Labs and allowed to acclimate for 2 weeks before experiments were performed.

## Tail vein metastasis model

Mice were restrained and tail vein injections were performed with 100 µl at $10^6$ cells per ml. Mice were not fed doxycycline—Dox (+) cells were treated for 24 hr before injections with 200 ng/ml in complete media. For bioluminescence imaging, mice were injected i.p. with D-luciferin at 15 mg/kg, anesthetized with 2.5% isofluorane and luciferin was allowed to circulate for 10 min prior to live animal imaging on an IVIS Spectum imaging system (PerkinElmer, Waltham, MA). Sample sizes were determined using a two-sample test, a power of 0.90, and a Type I error of 0.05 with data collected from pilot studies.

## Histology

Paraffin-embedded lung tissues were sectioned, deparaffinized, and stained with hematoxylin and eosin before imaging.

## Immunohistochemistry

Paraffin-embedded lung tissues were sectioned at 5 µm, deparaffinized, and rehydrated using graded ethanol washes. Sodium citrate buffer was used for antigen retrieval. Sections were blocked with BSA then incubated with primary antibodies overnight. Anti-mApple antibodies were from Abcam (Cambridge, UK) and used as recommended. Biotinylated secondary antibodies were from Cell Signaling Technology and used as recommended. DAB peroxidase substrate kit from Vector Labs (Burlingame, CA) was used as recommended. Sections were then dehydrated using graded ethanol washes and mounted in Permount (Sigma). Mounted sections were imaged by light microscopy.

## Immunofluorescence

Paraffin-embedded lung tissues were sectioned at 5 µm, deparaffinized, and rehydrated using graded ethanol washes. Sodium citrate buffer was used for antigen retrieval. Sections were blocked with BSA then incubated with primary antibodies overnight. pH3 antibodies were from Cell Signaling Technology and used as recommended. Fluorescent secondary antibodies were from Jackson ImmunoResearch (West Grove, PA) and used as recommended. Sections were wet-mounted then imaged by confocal microscopy.

## Quantification of immunofluorescence

After microscopy, images were analyzed using ImageJ software. DAPI-stained nuclei were counted using the 'Analyze particles' tool. Signal from antibodies of interest was measured using the 'Measure' tool set to output 'mean' pixel intensity. Quotients were obtained by dividing either the

number of positive nuclei (pH3) or mean pixel intensity (other antibodies) by the number of nuclei—thereby accounting for tumor density or size in our image analysis. Quotients were normalized to negative controls and plotted as normalized average signal per cell.

## Acknowledgements

The authors would like to thank Dr. Jon Lakins for the rtTA lentiviral vector. We would also like to thank Drs. Ori Maller, Dwight Chambers, and Brian Belardi for helpful discussions. This work was funded by a grant from the US National Institutes of Health (R01GM59907). ECW was supported by a predoctoral fellowship (F31CA200544).

## Additional information

### Funding

| Funder | Grant reference number | Author |
| --- | --- | --- |
| National Institute of General Medical Sciences | R01GM59907 | Valerie M Weaver<br>Carolyn R Bertozzi |
| National Cancer Institute | F31CA200544 | Elliot C Woods |

The funders had no role in study design, data collection and interpretation, or the decision to submit the work for publication.

### Author contributions

Elliot C Woods, Conceptualization, Data curation, Formal analysis, Supervision, Funding acquisition, Validation, Investigation, Visualization, Methodology, Writing—original draft, Writing—review and editing; FuiBoon Kai, Michael W Pickup, Data curation, Validation, Investigation, Methodology; J Matthew Barnes, Conceptualization, Data curation, Formal analysis, Validation, Investigation, Methodology, Writing—review and editing; Kayvon Pedram, Michael J Hollander, Data curation, Methodology, Writing—review and editing; Valerie M Weaver, Carolyn R Bertozzi, Conceptualization, Resources, Formal analysis, Supervision, Funding acquisition, Investigation, Visualization, Methodology, Writing—original draft, Project administration, Writing—review and editing

### Author ORCIDs

Elliot C Woods http://orcid.org/0000-0003-2920-8334
Kayvon Pedram https://orcid.org/0000-0001-8365-1826
Carolyn R Bertozzi http://orcid.org/0000-0003-4482-2754

### Ethics

Animal experimentation: The studies herein were performed in accordance with the guidelines in the Guide for the Care and Use of Laboratory Animals of the National Institutes of Health. All mice were maintained in accordance with University of California Institutional Animal Care and Use Committee guidelines under protocol AN105326. The protocol was also approved by the Administrative Panel on Laboratory Animal Care (APLAC) of Stanford University under protocol APLAC-30885.

### Decision letter and Author response

Decision letter https://doi.org/10.7554/eLife.25752.020
Author response https://doi.org/10.7554/eLife.25752.021

## Additional files

### Supplementary files

• Transparent reporting form
DOI: https://doi.org/10.7554/eLife.25752.017

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
