## [Decision Letter]

Thank you for submitting your article "A bulky glycocalyx fosters metastasis formation by promoting G1 cell cycle progression" for consideration by *eLife*. Your article has been reviewed by three peer reviewers, and the evaluation has been overseen by Johanna Ivaska as the Reviewing Editor and Philip Cole as the Senior Editor. The following individual involved in review of your submission has agreed to reveal his identity: Ron Schnaar (Reviewer #2).

The reviewers have discussed the reviews with one another and the Reviewing Editor has drafted this decision to help you prepare a revised submission.

Summary:

Aberrant expression of MUC1 is found in the majority of carcinomas and in hematological cancers. It has been demonstrated earlier by the authors of this manuscript that addition of ~80nm glycoprotein mimetics or enforced expression of MUC1 stimulates survival and proliferation of non-transformed MECs on soft substrates that would otherwise not support survival (Nature 2015). It was also found that oncogenic transformation, which allows these MECs to proliferate on soft substrates, is accompanied by increased expression of MUC1 on these cells. This earlier work indicated that a thicker glycocalyx augments growth-promoting integrin signaling through tension applied to ECM-bound integrins.

In the current manuscript, the authors extend their work and show that expression of a tailless MUC1 mutant or coating with a ~90nm synthetic GalcNAc polymer is sufficient to support outgrowth of IV-injected 4TO7 breast cancer cells in the mouse lung. This represents a new finding, demonstrating that manipulation of the glycocalyx at the extracellular surface, rather than signaling by MUCs drives metastatic outgrowth. Mechanistic studies in vitro revealing that MCF10A cells coated with large polymers show proliferation even on soft substrates is largely a confirmation of previous work (Nature 2015). The idea that FAK and PI3K-AKT signaling as well as cyclin D accumulation is increased on soft substrates and in metastatic lesions in the presence of the large polymers, extends the previous work but in its current form is not fully convincing.

Overall the reviewers find the work interesting, largely novel, and the data in the manuscript to be mainly of good quality. However, additional controls are required and further exploration of alternative explanations of the data should be considered. In addition, more Introduction/Discussion regarding the kinetic funneling concept are needed.

Essential revisions:

1) Figure 1 – controls are required showing what the metastatic potential of parental 4T07 cells are in this experimental set up. There is significant metastatic formation following treatment of the cells with the short polymer even though there is no signal activation in the in vitro model with this polymer. Therefore it is necessary to show whether this polymer is also having a significant effect in vivo.

2) Figure 1: It is important to rigorously rule out that the bulky glycocalyx in fact increases the propensity of IV injected tumor cells to get stuck in the lung vasculature. Figure 1 shows that the number of macroscopically detectable lung mets is higher in the presence of the long polymer. Is this in any way related to more cells getting trapped? For this, lung tissue can be isolated at early time points post injection (days 1, 2, 3, 4, 5) and the number of trapped tumor cells quantitatively determined by qPCR.

3) Figure 1, Figure 4 versus 2, 3: Why do the authors switch from 4TO7 to MCF10A here? The in vivo work is with 4TO7, the in vitro experiments with MCF10A. We agree that MCF10A is a good model but are the same results seen with 4TO7 cells? Are 4TO7 cells indeed not showing proliferation on soft substrates and is FAK, PI3K-AKT signaling, cyclin D accumulation in those cells under those conditions stimulated by the long polymers? Not all experiments, but at least confirmation of a few key findings across these two models would be required to directly relate in vitro work in Figure 2 and Figure 3 to the in vivo experiments in Figure 1 and 4.

4) How valid is it to compare the signaling events in the in vitro model with those seen in vivo? For example activation of AKT signaling was reported in vitro but not shown in vivo. pAKT IHC should be carried out on lung metastases in the experiments described for the polymer coated cells and also those where MUC1ΔCT-expressing cells were used. Only then can the authors claim that the PI3K/AKT pathway plays a role in the lung environment.

5) Figure 2 and Figure 3: Increased pH3 levels (Figure 2) and FAK-pY925 (Figure 3) on soft substrate coated with FN in presence of long polymers is clear. However, increased cyclin D levels (compared to PBS and short polymer in Figure 2) and increased AKT phosphorylation (compared to PBS control in Figure 2) are not convincing at all and increased FAK-pY397 is very modest at best (compared to PBS and short polymer in Figure 3). Western blots are indicated by the authors to be representative of at least 2 (Figure 2) or 3 (Figure 3) experiments. If this is the best example, this data is not convincing. Notably, the effects on pH3 and colony size are clear so the effect on proliferation is well-demonstrated. However, specifically for cyclin D, p-AKT, and p-Y397FAK data, the authors should present WB data where differences are convincing and rigorous quantification of 3 or more biologically independent experiments need to be provided.

6) pH3 analysis was carried out on 3-4 tumors per mouse. Does the level of positivity depend on the size of the individual tumors? As this analysis was carried out at a single time point it provides a snapshot of proliferation at that time. It may be that reduced proliferation in the short polymer treated cells reflects a delay in their initial growth (and therefore size of the individual tumors) and not one of reduced proliferation per se. The same argument could be made for the cyclin D1 analysis.

7) Figure 4: were the mice fed doxycline? What are the levels of MUC1ΔCT at the time of sacrifice? If the expression is not maintained for the duration of the experiment it could again suggest that the effects of MUC1ΔCT may be mediated at very early stages of tumor cell arrest and extravasation in the lung.

8) Cells were "painted.… with glycopolymers" but the methods used are not apparent within the manuscript. Details sufficient for reproducing this aspect of the experimental approach would be welcome.

[Editors' note: further revisions were requested prior to acceptance, as described below.]

Thank you for submitting your article "A bulky glycocalyx fosters metastasis formation by promoting G1 cell cycle progression" for consideration by *eLife*. Evaluation of your article has been overseen by a Reviewing Editor and Philip Cole as the Senior Editor.

After discussion, the Reviewing Editor has drafted this decision to help you prepare a revised submission.

Summary:

The authors have revised the manuscript considerably and addressed many of the previous points raised by the reviewers. However, it is felt that two of the main points need further work (see below).

Essential revisions:

"Point 3) Figure 1, Figure 4 versus 2, 3: Why do the authors switch from 4TO7 to MCF10A here? The in vivo work is with 4TO7, the in vitro experiments with MCF10A. We agree that MCF10A is a good model but are the same results seen with 4TO7 cells? Are 4TO7 cells indeed not showing proliferation on soft substrates and is FAK, PI3K-AKT signaling, cyclin D accumulation in those cells under those conditions stimulated by the long polymers? Not all experiments, but at least confirmation of a few key findings across these two models would be required to directly relate in vitro work in Figure 2 and Figure 3 to the in vivo experiments in Figure 1 and 4."

To their credit, the authors have explained why MCF10A cells were not used for the metastasis in vivo studies and have added data that show the limitations in this regard. However, it was not clear to the reviewers why 4T07 cells were not studied 'in vitro' in terms of cell signaling. It would seem that these cells can be analyzed in culture since the authors discuss growing them to sub-confluency. We thus ask the authors to investigate the FAK, PI3K-AKT, and cyclin D status in 4T07 as this would be important to demonstrate the generality and robustness of the main findings.

"Point 5) However, specifically for cyclin D, p-AKT, and p-Y397FAK data, the authors should present WB data where differences are convincing and rigorous quantification of 3 or more biologically independent experiments need to be provided."

We appreciate that the authors have now added densitometry and error bars to the western blot results in the revised manuscript. However, given the limitations and challenges of western blots, it is desirable to perform at least three replicates and calculate p values to determine the reliability and significance of the findings. We hope that the authors can therefore add at least one additional replicate to the experiments on cyclin D, pAKT and pY397FAK to ensure the robustness of the results.

---

## [Author Response]

Essential revisions:1) Figure 1 – controls are required showing what the metastatic potential of parental 4T07 cells are in this experimental set up. There is significant metastatic formation following treatment of the cells with the short polymer even though there is no signal activation in the in vitro model with this polymer. Therefore it is necessary to show whether this polymer is also having a significant effect in vivo.

The reviewers bring up a very important point: our negative control (short glycopolymer-bearing cells) requires validation. We have added a new supplementary figure to illustrate that validation: Figure 1—figure supplement 3. We can determine the effect of short glycopolymers on metastasis by comparing the metastatic potential of cells treated with these glycopolymers or merely mock treated with PBS. The negative control in Figure 4 – doxycycline withheld cells – is effectively mock-treated naïve 4TO7 cells, so we constructed this new figure by compiling the data from this control experiment with the one from Figure 1, which uses short glycopolymer-bearing cells. The figure demonstrates that short glycopolymer-bearing cells cause no increase in tumor burden as measured by quantitative bioluminescence imaging (A), lung mass (B), or number of frank mets (C) compared to PBS-treated cells. We have added reference to this new figure in the third paragraph of the subsection “The glycocalyx increases metastatic potential in a size dependent manner”.

2) Figure 1: It is important to rigorously rule out that the bulky glycocalyx in fact increases the propensity of IV injected tumor cells to get stuck in the lung vasculature. Figure 1 shows that the number of macroscopically detectable lung mets is higher in the presence of the long polymer. Is this in any way related to more cells getting trapped? For this, lung tissue can be isolated at early time points post injection (days 1, 2, 3, 4, 5) and the number of trapped tumor cells quantitatively determined by qPCR.

The reviewers bring up an important potential explanation. The experiments in the paper do actually address this question, but perhaps in a way that would not benefit the reader as originally written. The graphs in Figure 1—figure supplement 1 and Figure 4—figure supplement 1 show that we measured bioluminescence of the mice on day 0, just 4 hours after injection when, as seen in Figure 1, only those cells stuck in the lungs are left after clearing from the rest of the vasculature. By comparing photon flux at this time point, we can quantitatively measure the number of trapped tumor cells (since the mice themselves generate essentially zero background photons). The treatment groups show no significant increase in accumulated tumor cells relative to controls – see below. We have added a sentence explaining and hopefully making clear this finding in the first paragraph of the subsection “The glycocalyx increases metastatic potential in a size dependent manner”.

3) Figure 1, Figure 4 versus 2, 3: Why do the authors switch from 4TO7 to MCF10A here? The in vivo work is with 4TO7, the in vitro experiments with MCF10A. We agree that MCF10A is a good model but are the same results seen with 4TO7 cells? Are 4TO7 cells indeed not showing proliferation on soft substrates and is FAK, PI3K-AKT signaling, cyclin D accumulation in those cells under those conditions stimulated by the long polymers? Not all experiments, but at least confirmation of a few key findings across these two models would be required to directly relate in vitro work in Figure 2 and Figure 3 to the in vivo experiments in Figure 1 and 4.

To build upon the work in the 2014 Paszek et al.paper, in which we used MCF-10A cells as an in vitro model, we tested the ability of these same cells to seed metastases in our tail-vein-to-lung model. At early time points, we observed a delay in the onset of anchorage-dependent cell death based on glycocalyx size, just as in our in vitro model. Long glycopolymers seem to delay the onset of cell death as shown by the apoptosis marker cleaved caspase-3 (CC3). Being nontransformed cells, however, the MCF-10A cells were incapable of forming frank metastases. Eventually, the cells did die, likely because metastasis formation also relies upon additional factors including activation of an angiogenic response and immune cell evasion – two parameters extending well beyond the scope of the current proposal. This prevented us from being able to assess any effect the glycocalyx may have had on *proliferation* specifically.

Thus in order to study the effect of glycocalyx bulkiness on G1 cell cycle progression and cell proliferation, we moved to a new model – that of the syngeneic 4TO7/Balbc model. Upon tail vein injection, the cells in this model survive for extended periods of time in the lung. They fail, however, to proliferate and thus have been used extensively as a model of micrometastatic tumor dormancy. This dormancy "prone" tumor cell line, therefore, permitted us to specifically address the contribution of a bulky glycocalyx on cell proliferation and G1 cell cycle transit in vivo.

In our newly revised manuscript, we have outlined the rationale for the use of these different breast epithelial models including their complementary strengths/weaknesses and expanded the discussion of their use in the first paragraph of the Results. To demonstrate, as the reviewers requested, key findings of integrin activation across these two models, in our revised article we now include data from the aforementioned MCF-10A in vivo studies in a new supplementary figure, Figure 1—figure supplement 1. This important experiment not only validates critical in vitro findings in an in vivo setting, but the data also demonstrate the utility of the 4TO7/Balbc dormancy model, which permits a more direct exploration of the impact of the glycocalyx on cell proliferation in the metastatic settingin vivo.

4) How valid is it to compare the signaling events in the in vitro model with those seen in vivo? For example activation of AKT signaling was reported in vitro but not shown in vivo. pAKT IHC should be carried out on lung metastases in the experiments described for the polymer coated cells and also those where MUC1ΔCT-expressing cells were used. Only then can the authors claim that the PI3K/AKT pathway plays a role in the lung environment.

Indeed, the pAkt signaling should be apparent in the in vivo model as well. To address this, we performed IF-IHC on tissue sections from the experiments described in Figure 1 and 4. We have included these data as Figure 2—figure supplement 3 and Figure 4—figure supplement 2 as well as made reference to their inclusion in the fifth paragraph of the subsection “A thick glycocalyx drives cell cycle progression via the PI3K-Akt axis” and in the last paragraph of the subsection “The MUC1 ectodomain is sufficient to increase the metastatic potential of 4TO7 cancer cells”.

5) Figure 2 and Figure 3: Increased pH3 levels (Figure 2) and FAK-pY925 (Figure 3) on soft substrate coated with FN in presence of long polymers is clear. However, increased cyclin D levels (compared to PBS and short polymer in Figure 2) and increased AKT phosphorylation (compared to PBS control in Figure 2) are not convincing at all and increased FAK-pY397 is very modest at best (compared to PBS and short polymer in Figure 3). Western blots are indicated by the authors to be representative of at least 2 (Figure 2) or 3 (Figure 3) experiments. If this is the best example, this data is not convincing. Notably, the effects on pH3 and colony size are clear so the effect on proliferation is well-demonstrated. However, specifically for cyclin D, p-AKT, and p-Y397FAK data, the authors should present WB data where differences are convincing and rigorous quantification of 3 or more biologically independent experiments need to be provided.

As the reviewers have requested, we have quantified all of the western blot data for cyclin D1, pAkt and pFAKY397. The quantitative results confirm our subjective conclusions, and illustrate them in a more objective manner. This quantification was accomplished as follows: densitometry was performed on the blots, absolute intensity was normalized relative to the tubulin loading control and then normalized to the positive control lane (in each case, this was the band from cells loaded on stiff gels). Normalized values from replicate blots were averaged and are plotted in new supplementary figures as follows: cyclin D and pAkt were added as Figure 2—figure supplement 2, and pFAKY397 quantitation was added as Figure 3—figure supplement 1. We also added references to these new figures in the third paragraph of the subsection “A thick glycocalyx drives cell cycle progression via the PI3K-Akt axis” and in the first paragraph of the subsection “A thick glycocalyx stimulates integrin-FAK mechanosignaling”. These data bolster our in vivo findings illustrating the role of these markers in the tumor model.

6) pH3 analysis was carried out on 3-4 tumors per mouse. Does the level of positivity depend on the size of the individual tumors? As this analysis was carried out at a single time point it provides a snapshot of proliferation at that time. It may be that reduced proliferation in the short polymer treated cells reflects a delay in their initial growth (and therefore size of the individual tumors) and not one of reduced proliferation per se. The same argument could be made for the cyclin D1 analysis.

We have accounted for this possible bias in our image analysis. Image analysis software was used to quantify the number of nuclei present in the tumor in each image. The number of ‘positive’ pH3 nuclei – determined by a signal intensity threshold algorithmically in ImageJ – is then divided by the number of total nuclei. So the values are normalized for the number of nuclei present in the image (i.e. the size or density of the tumor) and they, therefore, do not affect our quantification. We have added clarification to this effect in the Materials and methods subsection “Quantification of immunofluorescence”.

7) Figure 4: were the mice fed doxycline? What are the levels of MUC1ΔCT at the time of sacrifice? If the expression is not maintained for the duration of the experiment it could again suggest that the effects of MUC1ΔCT may be mediated at very early stages of tumor cell arrest and extravasation in the lung.

The reviewers are correct that the mice are not fed doxycycline, and we have updated the Materials and methods section to make this more clear. The rational was that the glycopolymers, which emulate the effects of mucins such as MUC1, cannot be continuously administered, and so they also dilute out over time (and proliferative generations). While the effects of the MUC1 ectodomain may indeed be mediated at early stages of metastasis, the in vitro experiments demonstrating a proliferative phenotype were performed at these early time points with this in mind.

8) Cells were "painted.… with glycopolymers" but the methods used are not apparent within the manuscript. Details sufficient for reproducing this aspect of the experimental approach would be welcome.

The authors apologize for not making this clear in the original manuscript, since the glycopolymers play such an integral role to our experimentation. Their use was detailed in individual experimental Materials and methods sections, but we have added a new subsection to the Materials and methods entitled ‘Glycopolymer loading onto cell surfaces’ with the inclusion of reference to a paper describing both the synthesis and characterization of these glycopolymers. We hope this clarifies the procedure for readers.

[Editors' note: further revisions were requested prior to acceptance, as described below.]

Essential revisions:"Point 3) Figure 1, Figure 4 versus 2, 3: Why do the authors switch from 4TO7 to MCF10A here? The in vivo work is with 4TO7, the in vitro experiments with MCF10A. We agree that MCF10A is a good model but are the same results seen with 4TO7 cells? Are 4TO7 cells indeed not showing proliferation on soft substrates and is FAK, PI3K-AKT signaling, cyclin D accumulation in those cells under those conditions stimulated by the long polymers? Not all experiments, but at least confirmation of a few key findings across these two models would be required to directly relate in vitro work in Figure 2 and Figure 3 to the in vivo experiments in Figure 1 and 4."To their credit, the authors have explained why MCF10A cells were not used for the metastasis in vivo studies and have added data that show the limitations in this regard. However, it was not clear to the reviewers why 4T07 cells were not studied 'in vitro' in terms of cell signaling. It would seem that these cells can be analyzed in culture since the authors discuss growing them to sub-confluency. We thus ask the authors to investigate the FAK, PI3K-AKT, and cyclin D status in 4T07 as this would be important to demonstrate the generality and robustness of the main findings.

Indeed, 4TO7 cells can be analyzed in culture, and so we performed the experiments as requested above. 4TO7 cells were cultured, painted with long or short glycopolymers, then plated on soft (400 Pa) polyacrylamide gels with or without EGF challenge before being analyzed by immunoblotting. We found that, like MCF-10A cells, 4TO7 cells demonstrated increased FAK phosphorylation and increased Akt activation when long glycopolymers were used to bolster their glycocalyces when compared to short glycopolymers. Cyclin D1 expression levels, however, did not change in vitro. We posit that this is due to 4TO7 being a transformed cell line which gains only a marginal increase in cell-cycle driving forces with an increased glycocalyx in the rich environment of cell culture. This is demonstrated by our negative control conditions: cyclin D1 expression in PBS treated cells on soft gels. MCF-10A cells in these conditions demonstrate very little cyclin D1 expression, whereas 4TO7 cells show robust levels of cyclin D1 expression, even on soft gels with mock treatment (Figure 2—figure supplement 5). In addition, the nontransformed cell line MCF-10A does benefit from the increase in glycocalyx size in vitro, and our analysis of 4TO7 metastases in vivo demonstrates a glycocalyx length-dependency on cyclin D1 expression. For all of these results, we repeated the experiments at least thrice and quantified the results along with calculated p values. The data have been added as a combination of new panels in Figure 2 and Figure 3 as well as the new supplementary figure: Figure 2—figure supplement 5. Furthermore, the manuscript has been updated with references to these data in the subsection “A thick glycocalyx drives cell cycle progression via the PI3K-Akt axis”.

"Point 5) However, specifically for cyclin D, p-AKT, and p-Y397FAK data, the authors should present WB data where differences are convincing and rigorous quantification of 3 or more biologically independent experiments need to be provided."We appreciate that the authors have now added densitometry and error bars to the western blot results in the revised manuscript. However, given the limitations and challenges of western blots, it is desirable to perform at least three replicates and calculate p values to determine the reliability and significance of the findings. We hope that the authors can therefore add at least one additional replicate to the experiments on cyclin D, pAKT and pY397FAK to ensure the robustness of the results.

We have repeated these key results as requested so that the data reflect a minimum of three biological replicate experiments (n = 4 for pAkt and n = 5 for pFAK) for these three markers and have included calculated p values for each of the desitometric quantifications. Figure 2 and Figure 3 and the text in the subsection “A thick glycocalyx drives cell cycle progression via the PI3K-Akt axis” has been updated to reflect these added data in this revision.